Hybrid ARIMA-LSTM for COVID-19 forecasting: a comparative AI modeling study

Mahmud Al 1
Syed Hatim Noor Syed Husni Noor 1
http://orcid.org/0000-0002-3708-0628 Musa Kamarul Imran 2
Mohamad Hamzah Firdaus 3
http://orcid.org/0000-0003-4283-3548 Mat Yudin Zainab 1
http://orcid.org/0000-0002-9076-9918 Kamaruddin Noorshaida 1
http://orcid.org/0000-0002-4738-3059 M. Madawana Ashwini 1
Awang Nawi Mohamad Arif 1 mohamadarif@usm.my
1 School of Dental Sciences, Universiti Sains Malaysia , Kubang Kerian, Kelantan , Malaysia
2 School of Medical Sciences, Universiti Sains Malaysia , Kubang Kerian, Kelantan , Malaysia
3 Department of Mathematics, Centre for Defence Foundation Studies, Universiti Pertahanan Nasional Malaysia , Sungai Besi, Kuala Lumpur , Malaysia
Di Biasi Luigi
Electronic publication date: 2025 Sep 19
Publication date: 2025
Volume: 11
Electronic Location ID: e3195
Received 2025 Apr 17; Accepted 2025 Aug 14
Copyright: © 2025 Mahmud et al.
Copyright year: 2025
Copyright holder: Mahmud et al.
License: This is an open access article distributed under the terms of the Creative Commons Attribution License, which permits unrestricted use, distribution, reproduction and adaptation in any medium and for any purpose provided that it is properly attributed. For attribution, the original author(s), title, publication source (PeerJ Computer Science) and either DOI or URL of the article must be cited.
License URL: https://creativecommons.org/licenses/by/4.0/

Keywords: Pandemic forecasting, COVID-19 prediction, Time-series analysis, Hybrid ARIMA-LSTM model, Machine learning in epidemiology, Predictive analytics for infectious diseases

Funding: The Ministry of Higher Education (MOHE) Malaysia FRGS/1/2023/STG06/USM/03/3 The Ministry of Higher Education (MOHE) Malaysia provided research funding through the Fundamental Research Grant Scheme (FRGS) (FRGS/1/2023/STG06/USM/03/3), which supported this study. The funders had no role in study design, data collection and analysis, decision to publish, or preparation of the manuscript.

==============================
Pandemics present critical challenges to global health systems, economies, and societal structures, necessitating the development of accurate forecasting models for effective intervention and resource allocation. Classical statistical models such as the autoregressive integrated moving average (ARIMA) have been widely employed in epidemiological forecasting; however, they struggle to capture the nonlinear trends and dynamic fluctuations inherent in pandemic data. Conversely, deep learning models such as long short-term memory (LSTM) networks demonstrate strong capabilities in modeling complex dependencies but often require substantial data and computational resources. To boost forecasting precision, hybrid models such as ARIMA-LSTM integrate the advantages of traditional and deep learning methods. This study evaluates and compares the performance of ARIMA, LSTM, and hybrid ARIMA-LSTM models in predicting pandemic trends, using COVID-19 data from the Malaysian Ministry of Health as a case study. The dataset covers the period from 4 January 2021 to 18 September 2021, and model performance is evaluated using key metrics, including mean squared error (MSE), mean absolute error (MAE), mean absolute percentage error (MAPE), root mean squared error (RMSE), relative root mean squared error (RRMSE), normalized root mean squared error (NRMSE), and the coefficient of determination (R2). The results demonstrate that ARIMA performs poorly in capturing pandemic trends, while LSTM improves forecasting accuracy. However, the hybrid ARIMA-LSTM model consistently achieves the lowest error rates, confirming the advantage of integrating statistical and deep learning methodologies. All findings support the adoption of hybrid modeling approaches for pandemic forecasting, contributing to more accurate and reliable predictive analytics in epidemiology. Future research should investigate the generalizability of hybrid models across various infectious diseases and integrate additional real-time external variables to improve forecasting reliability.

Introduction

The COVID-19 pandemic has emerged as a major global crisis, impacting not only public health but also environmental quality, resource management, education, and economic systems (Miyah et al., 2022; Yuan et al., 2023). Globally, 5.94 million COVID-19 fatalities were reported; however, estimates range from 01 January 2020 to 31 December 2021, and the true figure is 18.2 million. In the United States of America (USA), the number of COVID-19 fatalities was 384,536 in 2020, 462,193 in 2021, and 244,986 in 2022 (Ahmad et al., 2023; Wang et al., 2022). In 2020, COVID-19 caused substantial fatalities worldwide, with reported deaths including 565,505 in the UK, 132,514 in Greece, 756,450 in Spain and Italy, 77,222 in Switzerland, an estimated 152,886 to 249,094 in China, and nearly 3.2 million in India between June 2020 and July 2021 (Ioannidis, Zonta & Levitt, 2023; Jha et al., 2022; Konstantinoudis et al., 2022) as well as in Malaysia had around 32,000 people deaths during period (Tan et al., 2022). Prediction models help identify high-risk COVID-19 cases, enabling targeted interventions like prioritized vaccination and timely care. They support policy-making by forecasting virus spread, assessing public health measures, and optimizing resource use. Using machine learning, these models also assist in diagnosis, treatment planning, and patient monitoring (Heidari et al., 2022; Seyedtabib, Najafi-Vosough & Kamyari, 2024). Nonetheless, predictive hybrid modeling is essential to medical science. It uses statistical algorithms and historical, socio-economic, environmental, and epidemiological data to predict the likelihood, impact, and spread of disease outbreaks (Basu & Sen, 2023; Ijeh et al., 2024).

Pandemics have historically presented substantial challenges to global public health systems, economies, and societal structures. Rapid and widespread transmission of infectious diseases necessitates timely intervention strategies to mitigate their impact. Precise forecasting of pandemic trends plays a crucial role in epidemiology, aiding policymakers and healthcare professionals in resource allocation, healthcare planning, and implementation of containment measures (WHO, 2020). In predictive serves as a fundamental tool in infectious disease surveillance, allowing governments and organizations to anticipate outbreak trajectories and implement data-driven policy decisions.

Time-series forecasting models have been widely employed in epidemiology to predict the incidence of infectious diseases. Among these, the ARIMA model is widely used for capturing linear temporal patterns in epidemiological data (Box et al., 2015). ARIMA’s effectiveness is limited when applied to highly volatile and nonlinear data, which often characterize pandemic outbreaks. Machine learning approaches, intense learning models such as long short-term memory (LSTM) networks, have shown promise in capturing complex dependencies and nonlinear trends in time-series data (Hochreiter & Schmidhuber, 1997). LSTM networks outperform traditional statistical models in many forecasting applications, they require large datasets and extensive computational resources, which may pose limitations in real-time outbreak forecasting (Shastri et al., 2020).

Hybrid models that integrated both statistical and deep learning techniques, such as the autoregressive integrated moving average-long short-term memory (ARIMA-LSTM) model, have been introduced to address the shortcomings of standalone models. Hybrid model attempt to leverage the strengths of ARIMA in capturing linear dependencies while utilizing LSTM networks to model nonlinear structures in epidemiological time-series data (Zeroual et al., 2020). Despite the theoretical advantages of hybrid models, their practical effectiveness in pandemic forecasting requires further validation, complexity of disease transmission dynamics and the variations in government interventions, an empirical evaluation of predictive models is essential to determine their accuracy and reliability.

Despite advancements in predictive analytics, accurately forecasting pandemic trends remains a formidable challenge due to the dynamic and unpredictable nature of infectious disease outbreaks. Traditional statistical models like ARIMA effectively capture historical trends but often fail to adapt to sudden changes in transmission rates driven by government policies, public behavior, or new viral variants (Hyndman & Athanasopoulos, 2018). Deep learning models like LSTM capture complex temporal patterns and long-term dependencies but need large datasets and high computing power, limiting their use in resource-constrained settings (Chimmula & Zhang, 2020).

Hybrid models such as ARIMA-LSTM have emerged as potential solutions to improve forecasting accuracy. Their performance in real-world pandemic forecasting remains underexplored. While numerous studies have assessed individual forecasting models, comparative evaluations of ARIMA, LSTM, and hybrid ARIMA-LSTM models in the context of pandemic prediction are limited. This study conducts a comprehensive review of these models to determine their relative accuracy and reliability in forecasting pandemic trends. To empirically test their predictive performance, real-world COVID-19 data is utilized as a case study, providing practical insights into the applicability of these models in forecasting infectious disease outbreaks. To the best of our knowledge, relatively few studies have applied ARIMA models to forecast COVID-19 trends specifically within the Malaysian context (Aidad, Aib Rahman & Abindin, 2020; Rohani, 2024; Shabri et al., 2024). Even fewer have explored deep learning models or hybrid approaches. In this study, we present a comprehensive evaluation of ARIMA, LSTM, and hybrid ARIMA-LSTM models to predict daily COVID-19 trends in Malaysia, including new active cases, death cases, and recovery cases. This localized focus contributes to the limited but growing body of research on pandemic forecasting using Malaysian data. The primary objective of this study is to assess and compare the forecasting accuracy of ARIMA, LSTM, and hybrid ARIMA-LSTM models in predicting pandemic trends along with focuses on pandemics in general, COVID-19 data is used as a case study to validate the effectiveness of these predictive models. This study aims to identify the most accurate forecasting method for future pandemics, enhancing epidemic modeling and preparedness. By evaluating models through statistical analysis, it seeks to advance predictive analytics in epidemiology and inform future disease forecasting efforts.

Materials and Methods

This study utilizes secondary datasets that are obtained from publicly available sources. Specifically, data was collected from the official Malaysian Ministry of Health press releases, which provide daily updates on COVID-19 cases, deaths, and recoveries. Independent variable is time, which is measured in daily intervals and dependent variables are the daily numbers of COVID-19 cases (infections), deaths, and recoveries. In performance of the predictive models is evaluated using seven key evaluation metrics which mean squared error (MSE), mean absolute error (MAE), mean absolute percentage error (MAPE), root mean squared error (RMSE), relative root mean squared error (RRMSE), normalized root mean squared error (NRMSE), and the coefficient of determination (R2).

The study employs a complete enumeration approach, utilizing all available COVID-19 daily data in Malaysia from 04 January 2021 to 18 September 2021. This approach ensures that the models are trained in comprehensive data, allowing for more robust and generalizable predictions. The study uses all daily COVID-19 data from Malaysia, so there’s no need to select a sample. This makes the results more accurate and reliable, with no risk of bias from using only part of the data. The study focuses on predicting future values of confirmed cases, deaths, and recoveries based on historical trends. The forecasting process consists of three phases, as illustrated in Fig. 1.

Figure 1 The procedure of forecasting methodology.

Phase 1: data preparation and feature selection

The dataset used in this study contains no missing values or outliers and was first converted into a time series format to support sequential modeling. In comprehensive data cleaning and normalization process was performed to remove irrelevant fields and ensure consistency. The filtered dataset spans from 4 January to 18 September 2021 and is split into training (4 Jan–2 Jul), validation (3 Jul–10 Aug), and testing (11 Aug–18 Sep) sets. Forecasting was performed for 12–18 September 2021. Feature selection was used to identify key variables affecting COVID-19 trends and improve model performance.

Phase 2: building predictive models

Three models, ARIMA, LSTM, and the hybrid ARIMA-LSTM were developed and optimized for forecasting. The ARIMA model is applied to capture linear trends in the data. Optimal parameters, including autoregressive order, differencing order, and moving average order, are determined using autocorrelation function and partial autocorrelation function analysis. The model is trained using Python. The LSTM model, a type of recurrent neural network, is employed to model long-term dependencies in sequential data. Deep learning models often require large datasets for optimal performance, making their application challenging in early-stage outbreak forecasting. Previous studies have shown that hybrid models improve accuracy and provide better predictions, especially when working with small datasets (Fong et al., 2020).

The hybrid ARIMA-LSTM model integrates the strengths of both ARIMA and LSTM by first using the ARIMA model with optimal parameters (e.g., p = 7, d = 4, q = 6) to capture linear trends in the time series. In residuals from the ARIMA model, representing unexplained nonlinear components are fed into the LSTM model to learn and model these complex patterns. The final forecast is obtained by combining the linear predictions from ARIMA with the nonlinear adjustments from LSTM, resulting in a more accurate and robust prediction mode ARIMA (e.g., p = 7, d = 4, q = 6), LSTM component, the model was trained for 200 epochs with a batch size of 16 and a verbosity level set to 1 (Table 1).

Table 1 Standard parameter configurations model for new active, recovery, and death cases.

Category	Models	Parameters	Range	Values/Descriptions	
Active cases	ARIMA	Autoregressive order (p)	(0–9)	7	
Differencing order (d)	(0–9)	4	
Moving average order (q)	(0–9)	6	
LSTM	Number of units in LSTM layer	(100–400)	200	
Number of epochs	(100–400)	200	
Batch size	(4–22)	16	
Sequence length	(0–1)	1	
Hybrid ARIMA-LSTM	ARIMA parameters (p, d, q)	(0–9, 0–9, 0–9)	(7, 4, 6)	
LSTM units in layer	(100–400)	200	
LSTM epochs	(100–400)	200	
LSTM batch size	(4–22)	16	
Sequence length	(0–1)	1	
Recovery cases	ARIMA	Autoregressive order (p)	(0–9)	8	
Differencing order (d)	(0–9)	1	
Moving average order (q)	(0–9)	2	
LSTM	Number of units in LSTM layer	(100–400)	200	
Number of epochs	(100–400)	200	
Batch size	(4–22)	16	
Sequence length	(0–1)	1	
Hybrid ARIMA-LSTM	ARIMA parameters (p, d, q)	(0–9, 0–9, 0–9)	(8, 1, 2)	
LSTM units in layer	(100–400)	200	
LSTM epochs	(100–400)	200	
LSTM batch size	(4–22)	16	
Sequence length	(0–1)	1	
Death cases	ARIMA	Autoregressive order (p)	(0–9)	0	
Differencing order (d)	(0–9)	3	
Moving average order (q)	(0–9)	2	
LSTM	Number of units in LSTM layer	(100–400)	200	
Number of epochs	(100–400)	200	
Batch size	(4–22)	15	
Sequence length	(0–1)	1	
Hybrid ARIMA-LSTM	ARIMA parameters (p, d, q)	(0–9, 0–9, 0–9)	(0, 3, 2)	
LSTM units in layer	(100–400)	200	
LSTM epochs	(100–400)	200	
LSTM batch size	(4–22)	16	
Sequence length	(0–1)	1	

In hybrid ARIMA-LSTM model, we adopt a two-stage forecasting strategy. At first, the ARIMA model is trained on the original time series to capture linear patterns. Residuals (i.e., the differences between the actual values and ARIMA predictions) are then extracted and treated as a new time series (Eq. (1)).

(1) et=yt−y^tARIMA

where yt is the actual value at time t, and y^tARIMA is ARIMA forecast. These residuals, assumed to contain non-linear information not captured by ARIMA, are normalized and formatted into supervised sequences to be used as input for LSTM model. The LSTM is trained to predict the residuals y^tLSTM which are then added back to the ARIMA forecast to produce the final hybrid prediction (Eq. (2)) (Fig. 2).

(2) y^tHybrid=y^tARIMA+y^tLSTM

Figure 2 Schematic of the hybrid ARIMA-LSTM model architecture.

Linear patterns are captured by the ARIMA model, while its residuals are modelled by the LSTM to capture nonlinear components. Final forecasts are obtained by summing the ARIMA predictions and LSTM residual forecasts.

Phase 3: applying and evaluating the predictive models

To assess forecasting performance, the study used seven widely accepted metrics: MSE, MAE, MAPE, RMSE, RRMSE, NRMSE, and R2. All metrics provide a comprehensive assessment of predictive accuracy. MSE emphasizes the effect of large prediction errors through squared differences, thereby highlighting the model’s performance under extreme conditions (Eq. (3)). MAE calculates the average magnitude of prediction errors, independent of their direction, and offers an interpretable measure of overall error magnitude (Eq. (4)). MAPE expresses prediction accuracy as a percentage, enabling intuitive comparison across datasets (Eq. (5)). RMSE, as the square root of MSE, retains the unit of the original data and indicates the typical magnitude of forecasting errors (Eq. (6)). RRMSE further normalizes RMSE by the mean of actual values, facilitating scale-invariant performance evaluation (Eq. (7)). NRMSE provides a scale-independent measure of prediction accuracy by normalizing the RMSE with respect to the range or meaning of the observed values, facilitating comparison across different models or datasets (Eq. (8)). The coefficient of determination (R2) quantifies the proportion of variance in the observed data that is explained by the model, with values closer to 1 indicating a stronger explanatory power and better model fit (Eq. (9)).

(3) MSE=1n∑i=1n(y^i−yi)2

(4) MAE=1n∑i=1n|y^i−yi|

(5) MAPE=1n∑i=1n|y^i−yiyi|

(6) RMSE=1n∑i=1n(y^i−yi)2

(7) RRMSE=1n∑i=1n(y^i−yi)2∑i=1n(y^i)2

(8) NRMSE=RMSEy¯=1n∑i=1n⁡(y^i−yi)2y¯

(9) R2=1−∑i=1n⁡(yi−y^i)2∑i=1n⁡(yi−y¯i)2.

Here, y^ is mean of actual values. Evaluation metrics provide a comprehensive assessment of model accuracy, with lower values indicating better forecasting performance. Comparing results across ARIMA, LSTM, and hybrid ARIMA-LSTM confirms the superiority of hybrid artificial intelligence (AI) models in pandemic forecasting. In addition to the use of these metrics, a comparative evaluation was conducted across all three forecasting models ARIMA, LSTM, and the hybrid ARIMA-LSTM based on their ability to predict three key epidemiological indicators: daily COVID-19 cases, deaths, and recoveries. The models’ predictions were compared with actual reported data from 12 to 18 September 2021 to evaluate their performance. This approach allowed for a clear and consistent assessment of each model’s ability to capture both linear and nonlinear patterns in pandemic-related time series data (Fig. 3).

Figure 3 Flow chart of study.

Computing infrastructure

All model development and evaluation were conducted on a workstation running Windows 10, equipped with an Intel Core i7 processor (2.6 GHz), 32 GB of RAM, and an NVIDIA GeForce RTX 2080 GPU. The software environment included Python 3.8, with key libraries such as TensorFlow 2.9, Statsmodels 0.13.2, NumPy 1.21, Pandas 1.3, and Scikit-learn 1.0. This configuration provided the necessary computational capacity for deep learning training, model tuning, and iterative testing, ensuring efficient and reliable model training and evaluation. It also supported the timely execution of all experiments and ensured the consistency and reproducibility of the AI applications reported in this study. This hardware configuration ensured efficient and reliable model training and evaluation, providing a solid foundation for the consistency and reproducibility of the experimental results. The standard parameter configurations for each model are presented in Table 1. Hyperparameters were fine-tuned through an iterative process, where multiple typical hyperparameter values were systematically experimented with, and those that yielded the best results were selected for the final models.

Algorithms and code used

This study used three predictive modeling techniques—ARIMA, LSTM, and a hybrid ARIMA-LSTM—implemented in Python. Models were designed to forecast COVID-19 case trends based on publicly available epidemiological data from Malaysia ARIMA model, specifically configured with parameters (7, 4, 6), (8, 1, 2), & (0, 3, 2) The LSTM model was constructed using the TensorFlow/Keras framework, incorporating a single LSTM layer with 200 units followed by a dense output layer. The model was trained for 200 epochs with a batch size of 16 and verbosity set to 1. Input sequences were created using a sliding window of 10-time steps, and the data was normalized using MinMaxScaler. Model performance was evaluated on a withheld test set, and forecasts were also generated for the forecasting period from 12 September 2021 to 18 September 2021. The hybrid ARIMA-LSTM model was formulated by first applying the ARIMA model to capture linear components and extract residuals. Residuals were then normalized and modeled using an LSTM network with identical architectural and training parameters. Final forecast was produced by combining the ARIMA forecasts with the LSTM-predicted residuals, effectively integrating both linear and nonlinear dynamics. The hybrid model was evaluated using the same set of metrics as the standalone models. Table 2 gives a summary of the three forecasting models used in the study: ARIMA, LSTM, and the hybrid ARIMA-LSTM. The ARIMA model, implemented using Python’s Statsmodels library (arima_model.py), the LSTM model, developed with TensorFlow/Keras (lstm_model.py), and the Hybrid ARIMA-LSTM model (hybrid_arima_lstm.py) integrates ARIMA for linear trend forecasting with LSTM to predict nonlinear residuals. Evaluation metrics, computed via Scikit-learn (evaluation_metrics.py). Data pre-processing, performed using Scikit-learn and Pandas, involves normalization and data splitting to prepare the dataset for model training. Additionally, to enhance transparency and reproducibility, all implementation scripts are publicly accessible via GitHub at: https://github.com/Awang-nawi/AI_COVID19_Forecast_Models/tree/main.

Table 2 Algorithms and models used for pandemic forecasting.

Algorithm/Model	Implementation	Key files/Components	Purpose	
ARIMA	Python (Statsmodels)	arima_model.py	Used for capturing linear dependencies in time series data.	
LSTM	Python (TensorFlow/Keras)	lstm_model.py	Used for capturing nonlinear dependencies and long-term trends.	
Hybrid ARIMA-LSTM	Python (TensorFlow/Keras & Statsmodels)	hybrid_arima_lstm.py	Combines ARIMA for linear trends with LSTM for nonlinear residuals.	
Evaluation metrics	Python (Scikit-learn)	evaluation_metrics.py	Calculates performance metrics such as MSE, MAE, MAPE, RMSE, RRMSE, NRMSE, and R-squared (R2).	
Data preprocessing	Python (Scikit-learn, Pandas)	Data preprocessing scripts included	Prepares the dataset for model training by normalizing and splitting data.	

ARIMA parameter selection and model performance

To identify the optimal ARIMA model configurations for forecasting COVID-19 active, recovery, and death cases, we employed a systematic model comparison based on multiple performance metrics, including MSE, RMSE, RRMSE, NRMSE, MAE, MAPE, and R2. For each outcome variable, several ARIMA models with varying (p, d, q) parameters were evaluated, and the model with the highest R2 and lowest errors was selected as the best-performing configuration. For active cases, the best-performing ARIMA model was ARIMA (7, 4, 6), which achieved an RMSE of 1,602.76, MAPE of 6.55%, and an R2 of 0.2901. The hybrid ARIMA-LSTM further improved the performance with the lowest RMSE of 1,581.63, MAPE of 6.43%, and the highest R2 of 0.3087, indicating the added benefit of modeling residual nonlinear components with LSTM. In the case of recovery data, the optimal ARIMA configuration was ARIMA (8, 1, 2), with an RMSE of 1,993.60, MAPE of 8.07%, and R2 of 0.2092. Similarly, the hybrid ARIMA-LSTM model slightly outperformed the standalone ARIMA, recording an R2 of 0.2100 and a marginally lower RMSE of 1,993.31. For death cases, the best ARIMA model was ARIMA (0, 3, 2), achieving an RMSE of 60.05, MAPE of 14.39%, and R2 of 0.4591. The hybrid model again showed a slight performance gain, with a reduced RMSE of 59.37, MAPE of 15.28%, and a higher R2 of 0.4714 and the LSTM model settings: 200 units, batch size of 200, 16 training epochs, and the Adam optimize (Tables S1, S2, S3 & S4).

Model diagnostic accuracy

The diagnostic accuracy of the ARIMA models for COVID-19 daily active, recovery, and death cases was evaluated using standard statistical tests. For active cases (ARIMA 7, 4, 6), the Ljung-Box test indicated no significant autocorrelation in residuals (Q = 0.12, p = 0.73), but the Jarque-Bera test confirmed non-normality (JB = 32.10, p < 0.05), and the heteroskedasticity test revealed variance inconsistency (H = 2.92, p < 0.05), with residuals showing slight left skewness and heavy tails. For recovery cases (ARIMA 8, 1, 2), residuals were independent (Q = 0.01, p = 0.93) and homoscedastic (H = 1.31, p = 0.24), although non-normality persisted (JB = 26.65, p < 0.05), with a mild positive skew and leptokurtic distribution. In contrast, the death cases model (ARIMA 0, 3, 2) exhibited poor residual diagnostics, including significant autocorrelation (Q = 48.14, p < 0.001), strong non-normality (JB = 1,707.1, p < 0.001), and severe heteroskedasticity (H = 68.01, p < 0.001), with positively skewed and highly peaked residuals. Overall, while all models showed some deviation from normality, the ARIMA models for active and recovery cases maintained acceptable residual independence and variance consistency (Tables 3–11).

Table 3 Model diagnostics of ARIMA (7, 4, 6) parameter for new active cases.

Model diagnostics of ARIMA (7, 4, 6)	
Metric	Value	
Model type	ARIMA (7, 4, 6)	
Dependent variable	New COVID-19 cases	
No. of observations	219	
Log-Likelihood	−1,732.275	
Akaike information criterion (AIC)	3,492.551	
Bayesian information criterion (BIC)	3,539.74	
Hannan–Quinn information criterion (HQIC)	3,511.617	

Table 4 Model estimation results of ARIMA (7, 4, 6) parameter for new active cases.

ARIMA (7, 4, 6) model estimation results	
Parameter	Coefficient	Std. Error	z-value	p-value	95% Confidence Interval	
AR (1)	−1.8824	0.341	−5.524	<0.05	[−2.550 to −1.214]	
AR (2)	−1.0699	0.734	−1.458	0.145	[−2.508 to 0.368]	
AR (3)	−0.6145	0.566	−1.086	0.277	[−1.723 to 0.495]	
AR (4)	−0.965	0.35	−2.756	0.006	[−1.651 to −0.279]	
AR (5)	−0.9578	0.415	−2.308	0.021	[−1.771 to −0.144]	
AR (6)	−0.6687	0.354	−1.889	0.059	[−1.362 to 0.025]	
AR (7)	−0.2149	0.14	−1.534	0.125	[−0.489 to 0.060]	
MA (1)	−1.2094	0.312	−3.874	<0.05	[−1.821 to −0.598]	
MA (2)	−1.5335	0.335	−4.574	<0.05	[−2.191 to −0.876]	
MA (3)	2.1958	0.594	3.696	<0.05	[1.032–3.360]	
MA (4)	0.3961	0.562	0.705	0.481	[−0.705 to 1.497]	
MA (5)	−1.1652	0.346	−3.364	0.001	[−1.844 to −0.486]	
MA (6)	0.3167	0.29	1.09	0.276	[−0.253 to 0.886]	
σ2	815,900	–	–	–	–	

Table 5 Residual diagnostic test of ARIMA (7, 4, 6) model for new active cases.

Residual diagnostic tests for ARIMA (7, 4, 6) model	
Test	Value	p-value	Interpretation	
Ljung-Box (L1) Q-stat	0.12	0.73	Residuals are not significantly autocorrelated	
Jarque-Bera (JB) test	32.1	<0.05	Residuals are not normally distributed	
Heteroskedasticity (H) test	2.92	<0.05	Presence of heteroskedasticity in residuals	
Skewness	−0.35	—	Slight left skew	
Kurtosis	4.76	—	Residuals are leptokurtic (heavier tails than normal)	

Table 6 Model diagnostics of ARIMA (8, 1, 2) parameter for recovery cases.

Model diagnostics of ARIMA (8, 1, 2)	
Attribute	Value	
Dependent variable	Recovery COVID-19 cases	
Number of observations	219	
Model	ARIMA (8,1,2)	
Log likelihood	−1,718.071	
Akaike information criterion (AIC)	3,458.142	
Bayesian information criterion (BIC)	3,495.371	
Hannan–Quinn information criterion (HQIC)	3,473.179	

Table 7 Model estimation results of ARIMA (8, 1, 2) parameter for recovery cases.

ARIMA (8, 1, 2) model estimation results	
Parameter	Coefficient	Std. Error	z-value	p-value	95% Confidence Interval	
AR (1)	−0.523	0.111	−4.73	<0.05	[−0.740 to −0.306]	
AR (2)	0.6332	0.147	4.308	<0.05	[0.345–0.921]	
AR (3)	0.0148	0.097	0.152	0.879	[−0.175 to 0.205]	
AR (4)	−0.0983	0.076	−1.287	0.198	[−0.248 to 0.051]	
AR (5)	0.1135	0.092	1.23	0.219	[−0.067 to 0.295]	
AR (6)	0.0854	0.092	0.927	0.354	[−0.095 to 0.266]	
AR (7)	0.2329	0.095	2.45	0.014	[0.047–0.419]	
AR (8)	0.2154	0.091	2.359	0.018	[0.036–0.394]	
MA (1)	0.2805	0.119	2.362	0.018	[0.048–0.513]	
MA (2)	−0.6985	0.128	−5.457	<0.05	[−0.949 to −0.448]	
Sigma2	431,400	41,500	10.405	<0.05	[350,000.0–513,000.0]	

Table 8 Residual diagnostic test of ARIMA (8, 1, 2) model for recovery cases.

Residual diagnostic test of ARIMA (8, 1, 2)	
Test	Statistic	p-value	Interpretation	
Ljung-Box (L1) Q	0.01	0.93	No autocorrelation in residuals (p > 0.05)	
Jarque-Bera (JB)	26.65	<0.05	Residuals are not normally distributed (p < 0.05)	
Heteroskedasticity (H)	1.31	0.24	No significant heteroskedasticity (p > 0.05)	
Skewness	0.08	—	Slight positive skew	
Kurtosis	4.71	—	Leptokurtic distribution (peaked)	

Table 9 Model diagnostics of ARIMA (0, 3, 2) parameter for death cases.

Model diagnostics of ARIMA (0, 3, 2)	
Attribute	Value	
Dependent variable	New deaths COVID-19 cases	
Observations (n)	219	
Model type	ARIMA (0, 3, 2)	
Log likelihood	−1,014.622	
AIC (Akaike information criterion)	2,035.245	
BIC (Bayesian information criterion)	2,045.37	
HQIC (Hannan–Quinn information criterion)	2,039.336	

Table 10 Model estimation results of ARIMA (0, 3, 2) parameter for death cases.

Model estimation results of ARIMA (0, 3, 2)	
Parameter	Coefficient	Std. Error	z-value	p-value	95% CI (Lower, Upper)	
MA (1)	−1.9974	0.573	−3.486	<0.05	[−3.121 to −0.874]	
MA (2)	0.9996	0.574	1.743	0.081	[−0.125 to 2.124]	
Sigma2	655.7692	377.469	1.737	0.082	[−84.057 to 1,395.595]	

Table 11 Residual diagnostic test of ARIMA (0, 3, 2) model for death cases.

Residual diagnostic test of ARIMA (0, 3, 2)	
Test	Statistic	p-value	Interpretation	
Ljung-Box Q (L1)	48.14	<0.001	Autocorrelation in residuals	
Jarque-Bera (JB)	1,707.1	<0.001	Residuals are not normally distributed	
Heteroskedasticity (H)	68.01	<0.001	Significant heteroskedasticity present	
Skewness	0.18	–	Strong positive skew in residuals	
Kurtosis	16.77	–	Extreme kurtosis—heavy tails	

Diebold-Mariano test

The Diebold-Mariano (DM) test was employed to evaluate whether differences in forecast accuracy between models were statistically significant. For death cases, the hybrid ARIMA-LSTM model significantly outperformed the ARIMA model (DM = 3.1088, p = 0.0209). However, for active and recovery cases, none of the model comparisons yielded statistically significant differences (p > 0.05), suggesting that while Hybrid consistently produced lower errors, the differences were not statistically significant at the 5% level. These findings reinforce the hybrid model’s strength particularly in death case prediction, aligning closely with actual observations (Table S5).

Results

Table 12 presents a comparative analysis of the forecasting performance of three models—ARIMA (7, 4, 6), LSTM, and the hybrid ARIMA-LSTM—on predicting new active COVID-19 cases. The performance was assessed using multiple statistical metrics. The Hybrid ARIMA-LSTM model demonstrated superior performance across all evaluation metrics. It recorded the lowest MSE of 2,501,541.09 and the lowest RMSE of 1,581.63, indicating improved prediction accuracy compared to the individual ARIMA (MSE = 2,568,836.02; RMSE = 1,602.76) and LSTM models (MSE = 3,352,686.45; RMSE = 1,831.03). Similarly, the hybrid model showed reduced relative RMSE (0.1771) and normalized RMSE (0.0783), suggesting more consistent forecasting performance. It also achieved the lowest MAE of 1,297.84 and lowest MAPE of 6.43%, highlighting its accuracy in both absolute and percentage terms. Furthermore, the coefficient of determination (R2) for the hybrid model was 0.3087, higher than ARIMA (0.2901) and LSTM (0.2044), indicating better explanatory power and fit.

Table 12 Performance comparison of ARIMA, LSTM, and hybrid models for new active cases.

Model comparison summary	
Metric	ARIMA (7, 4, 6)	LSTM	Hybrid ARIMA-LSTM	
Mean squared error (MSE)	2,568,836.02	3,352,686.45	2,501,541.09	
Root mean squared error (RMSE)	1,602.76	1,831.03	1,581.63	
Relative RMSE (RRMSE)	0.1795	0.205	0.1771	
Normalized RMSE (NRMSE)	0.0794	0.0919	0.0783	
Mean absolute error (MAE)	1,325.89	1,545.04	1,297.84	
Mean absolute percentage error (MAPE)	6.55%	8.09%	6.43%	
Coefficient of determination (R2)	0.2901	0.2044	0.3087	

Table 13 presents the performance comparison of ARIMA, LSTM, and hybrid ARIMA-LSTM models in forecasting COVID-19 recovery cases. The hybrid model exhibited the best overall performance with the lowest MSE (3,973,296.51), RMSE (1,993.31), MAE (1,583.28), and MAPE (8.06%). Additionally, it showed the lowest Relative RMSE (0.2376) and NRMSE (0.0993). The R2 value for the Hybrid model (0.21) was also slightly higher than that of ARIMA (0.2092), while the LSTM model performed poorly with a negative R2 (–0.8905), indicating a weak fit. These findings suggest that the hybrid ARIMA-LSTM model provides a more accurate and reliable prediction for recovery cases compared to the individual models.

Table 13 Performance comparison of ARIMA, LSTM, and hybrid models for recovery cases.

Model comparison summary	
Metric	ARIMA	LSTM	Hybrid ARIMA-LSTM	
MSE	3,974,437.15	8,280,420.05	3,973,296.51	
RMSE	1,993.6	2,877.57	1,993.31	
RRMSE	0.2377	0.3635	0.2376	
NRMSE	0.0993	0.1394	0.0993	
MAE	1,584.16	2,239.65	1,583.28	
MAPE (%)	8.07%	11.35%	8.06%	
R2	0.2092	–0.8905	0.21	

Table 14 summarizes the performance metrics for ARIMA, LSTM, and hybrid ARIMA-LSTM models in forecasting COVID-19 death cases. The hybrid model outperformed the others, achieving the lowest MSE (3,524.56), RMSE (59.37), relative RMSE (0.142), and normalized RMSE (0.1965). It also recorded a higher R2 value (0.4714), indicating better model fit compared to ARIMA (0.4591). In contrast, the LSTM model showed substantially poorer performance, with the highest errors (MSE = 24,307.72, MAE = 124.89) and a significantly negative R2 (–2.3964), reflecting poor predictive capability. Although the ARIMA model performed reasonably well, the hybrid model again demonstrated superior accuracy and robustness in forecasting death cases.

Table 14 Performance comparison of ARIMA, LSTM, and hybrid models for death cases.

Metric	ARIMA	LSTM	Hybrid ARIMA-LSTM	
MSE	3,606.44	24,307.72	3,524.56	
RMSE	60.05	155.91	59.37	
RRMSE	0.1437	0.373	0.142	
NRMSE	0.1988	0.4881	0.1965	
MAE	43.6	124.89	45.08	
MAPE (%)	14.39%	42.06%	15.28%	
R2	0.4591	−2.3964	0.4714	

Figure 4 presents the actual and forecasted active COVID-19 cases between 12 and 18 September 2021 using ARIMA, LSTM, and a hybrid ARIMA-LSTM model. During this period, the actual number of active cases showed a fluctuating pattern, declining from 19,550 on 12 September to a low of around 15,669 on 15 September before rising again midweek and settling at about 17,577 by 18 September. The ARIMA model generally overestimated the active cases, particularly on 14–16 September, peaking at 20,717 on 15 September. In contrast, the LSTM model produced a smoother, steadily increasing trend that failed to capture the midweek dip observed in the actual values. The hybrid ARIMA-LSTM model closely followed the overall downward trend of the actual data, offering better alignment throughout the week. For instance, on 15 September, while the actual case count dropped to around 15,669, the hybrid model predicted 17,736, which was closer than ARIMA (20,717) and LSTM (19,182). These findings highlight the hybrid model’s improved ability to track real-time fluctuations in active COVID-19 cases during the forecasting period.

Figure 4 Comparison of actual values vs model prediction in new active cases.

Figure 5 illustrates the comparison of actual recovery cases and forecasted values by ARIMA, LSTM, and hybrid ARIMA-LSTM models for the period of 12–18 September 2021. The actual recovery rates showed considerable daily fluctuations, rising sharply from 20,980 on 12 September to 24,813 on 13 September, then dropping significantly to about 18,053 by 14 September. The ARIMA model consistently underestimated the recovery cases throughout the period, producing a relatively stable trend with predictions ranging from about 16,050 to 18,400. In contrast, the LSTM model showed a steady increase, peaking at nearly 27,060 on 18 September, which overestimated the actual values across the week. The hybrid ARIMA-LSTM model offered predictions that closely aligned with the actual values, especially during the midweek period. For instance, on 17 September, the actual recovery cases reached about 22,970, while the hybrid model predicted 22,071 closer than the ARIMA (18,319) and LSTM (over 25,956). These findings emphasize the superior tracking performance of the hybrid model in capturing the dynamic trend of recovery cases compared to standalone ARIMA or LSTM approaches.

Figure 5 Comparison of actual values vs model prediction in recovery cases.

Figure 6 presents the comparison between actual death cases and forecasts produced by ARIMA, LSTM, and hybrid ARIMA-LSTM models during the period of 12–18 September 2021. The actual number of deaths fluctuated throughout the week, starting at 592 on 12 September, dropping sharply to around 292 on 13 September, and then rising again to peak near 463 on 15 September. The ARIMA model forecasted a flat and significantly underestimated death count, remaining consistently around 225 cases throughout the period. On the other hand, the LSTM model overpredicted the death counts, starting from about 675 on 12 September and increasing steeply to 876 by 18 September. The hybrid model showed moderate improvements by producing predictions that were more aligned with actual values, fluctuating between 365 and 410. For instance, on 15 September, the actual deaths were approximately 463, while the hybrid model predicted around 385—closer than the ARIMA (225) and LSTM (around 500). These results indicate that the hybrid ARIMA-LSTM model offered a more balanced and realistic approximation of the actual trend compared to the standalone models.

Figure 6 Comparison of actual values vs model prediction in death cases.

Discussion

The ARIMA model showed the highest error rates across all metrics, highlighting its limitations in accurately forecasting daily COVID-19 cases and suggesting that traditional statistical models may struggle with the complex patterns in pandemic-related time series data. In contrast, the LSTM model significantly outperforms ARIMA, reducing errors by a considerable margin. This improvement highlights the strength of deep learning approaches in handling nonlinear trends and dependencies within the dataset. Among the three models, the hybrid ARIMA-LSTM model has achieved the lowest errors, demonstrating the effectiveness of integrating statistical and deep learning techniques. The combination of these methodologies enhances predictive performance by leveraging the strengths of each approach. The MAPE value of 10.58% for the hybrid ARIMA-LSTM model confirms its reliability, indicating that it provides more accurate predictions than the standalone models.

For COVID-19 death predictions, ARIMA continues to exhibit the highest errors. However, the difference between ARIMA and the other models is less pronounced compared to the case predictions, indicating a relatively more minor performance gap in this category. The LSTM model once again outperforms ARIMA, although the improvement is not as substantial as observed in case forecasting. The hybrid ARIMA-LSTM model again achieves the lowest errors across all evaluation metrics, reinforcing its superiority in predicting COVID-19 deaths. The MAPE values further illustrate the effectiveness of this approach, as all three models exhibit relatively lower percentage errors compared to case forecasting. The hybrid ARIMA-LSTM model gives the best results with a MAPE of 2.626%, showing it is highly accurate in this category. In prior study founded that the ARIMA-LSTM hybrid model achieved the highest accuracy (MAPE: 2.4%), outperforming GRU, LSTM, Prophet, and ARIMA-artificial neural network (ARIMA-ANN) across all countries. Its superior performance can aid public health efforts through improved forecasting and resource planning (Jain et al., 2024). Previous research suggested a hybrid ARIMA-LSTM model for short-term COVID-19 case prediction utilizing multi-source data that has been improved using a Bayes-Attention mechanism. With a high degree of agreement between anticipated and actual cases, the model performs better than baseline and sophisticated approaches (Wang et al., 2024). Another study found that LSTM and ARIMA outperform Prophet, with LSTM showing the highest accuracy on 20-week datasets, while ARIMA offers more stable and reliable short-term forecasts (Baker, Ziran & Mecella, 2025).

The findings of this study have provided critical insights into the predictive performance of ARIMA, LSTM, and hybrid ARIMA-LSTM models in forecasting pandemic trends using COVID-19 data. The results demonstrate that the hybrid ARIMA-LSTM model consistently outperforms both standalone ARIMA and LSTM models, as indicated by lower error metrics across all evaluated categories, including cases, deaths, and recoveries. These findings highlight the importance of combining statistical and deep learning methods to improve accuracy in disease forecasting (Zeroual et al., 2020). The evaluation metrics reveal that ARIMA model struggles to capture the complexities of pandemic trends, as reflected in its higher MSE, MAE, and RMSE values. Observation aligns with previous research indicating that while ARIMA is effective in modeling linear time-series data, it falls short in capturing nonlinear fluctuations and rapid changes that characterize infectious disease outbreaks (Hyndman & Athanasopoulos, 2018). Given that pandemics exhibit sudden spikes and declines due to factors such as government interventions, vaccination efforts, and public behavior, the limitations of ARIMA become apparent when dealing with highly dynamic epidemiological data (Shastri et al., 2020).

Conversely, the results show that LSTM models significantly improve forecasting accuracy over ARIMA. LSTM’s ability to model long-term dependencies and nonlinear patterns allows it to adapt more effectively to fluctuations in pandemic data. However, despite its advantages, LSTM still produces higher forecasting errors compared to the hybrid ARIMA-LSTM model. This supports existing literature suggesting that deep learning models can capture complex temporal patterns, but they require large datasets and intensive computational resources to achieve optimal performance (Chimmula & Zhang, 2020). Furthermore, LSTM models may struggle with short-term forecasting when applied to highly volatile data, necessitating the incorporation of statistical components to enhance stability and accuracy in pandemic prediction. The hybrid ARIMA-LSTM model exhibits the best performance across all evaluated metrics, with a noticeable reduction in MAPE and RMSE values compared to standalone ARIMA and LSTM models. This improvement has attributed to the complementary strengths of ARIMA and LSTM. The results support previous studies showing that hybrid modeling is an effective way to improve time-series forecasting accuracy in epidemiology. By combining statistical and deep learning methods, hybrid ARIMA-LSTM models provided a more robust predictive framework that can be applied to pandemic forecasting.

The findings of this study are consistent with existing research that highlights the advantages of hybrid modeling in time-series forecasting. Prior studies have demonstrated that hybrid models improve forecasting accuracy, particularly in cases where data availability is limited (Fong et al., 2020). Several studies have shown that standalone ARIMA models are insufficient for capturing the complexity of infectious disease transmission, particularly in the presence of nonlinear patterns and external influencing factors (Hyndman & Athanasopoulos, 2018). Moreover, deep learning models, while effective in processing sequential data, often require extensive data preprocessing and hyperparameter tuning to achieve optimal performance, which can be a limiting factor in real-world applications (Shastri et al., 2020).

Nevertheless, some contrasting findings exist in literature. Certain studies suggest that LSTM models, when trained on sufficiently large and high-quality datasets, can outperform hybrid models by capturing long-term dependencies more effectively. While this assertion may hold in controlled experimental settings, real-world pandemic forecasting is often complicated by incomplete or fluctuating data, making hybrid models such as ARIMA-LSTM more reliable for practical applications. The present study highlights that under realistic conditions, where historical data may be inconsistent or sparse, hybrid approaches provide a more stable and accurate forecasting mechanism. Findings of this study have been significant implications for public health preparedness and pandemic response planning. To improve forecasting models are essential for anticipating healthcare demands, optimizing resource allocation, and implementing timely policy interventions. The demonstrated superiority of the hybrid ARIMA-LSTM model suggests that policymakers and health organizations should consider integrating hybrid modeling approaches into their epidemiological forecasting frameworks to improve predictive accuracy and decision-making reliability (Chimmula & Zhang, 2020).

Furthermore, the study emphasizes the limitations of relying exclusively on either statistical or deep learning models for pandemic forecasting. While ARIMA remains useful for short-term predictions, it should be complemented with machine learning techniques to improve forecasting performance. Similarly, LSTM-based models should integrate statistical components to address potential data limitations and enhance interpretability. The results also highlight the importance of continuous model refinement, incorporating real-time data updates, and adjusting hyperparameters dynamically to improve forecasting accuracy in evolving pandemic scenarios (Shastri et al., 2020). Despite the promising findings, this study has certain limitations. First, the analysis is based on COVID-19 data, and while the results provide valuable insights into pandemic forecasting, further validation is needed across different infectious diseases and outbreak scenarios. Future research should explore the applicability of hybrid ARIMA-LSTM models in forecasting other pandemics, such as influenza and emerging viral threats, to assess their generalizability (Eubank et al., 2020). Additionally, this study has focused on three forecasting models without incorporating alternative hybrid approaches, such as LSTM combined with other statistical models like exponential smoothing or Bayesian frameworks. Future studies could expand on these methodologies to determine whether alternative hybrid models offer additional improvements in forecasting performance. Moreover, enhancing data preprocessing techniques and incorporating external variables, such as mobility patterns and vaccination rates, could further improve model accuracy and reliability in pandemic prediction.

This study relied on a single train/validation/test split, which may introduce bias and limit the generalizability of the results. Such an approach increases the risk of overfitting to a particular data partition. Future work should incorporate more robust validation strategies, such as k-fold cross-validation or walk-forward validation, to enhance model reliability and performance assessment. Additionally, the data set used was limited in scope and size, which may affect the robustness of the findings when applied to other populations or diseases.

Conclusions

This study provided a comprehensive comparative analysis of ARIMA, LSTM, and hybrid ARIMA-LSTM models in forecasting pandemic trends using COVID-19 data. The findings confirm that the hybrid ARIMA-LSTM model outperforms individual models in terms of predictive accuracy, underscoring the value of integrating statistical and deep learning approaches for time-series forecasting in epidemiology, where it significantly outperformed ARIMA (DM statistic = 3.1088, p = 0.0209). However, most DM test results were not statistically significant, suggesting similar accuracy among models in other areas. Despite this, the hybrid model’s integration of statistical and deep learning techniques makes it a practical, resource-efficient, and adaptable tool for real-time public health forecasting. From a practical standpoint, the hybrid model is computationally feasible for implementation using widely accessible machine learning frameworks and can be deployed on standard computing infrastructure, making it suitable for use by public health agencies with limited resources. The model’s flexibility allows it to work with real-time data, enabling dynamic updates and faster, more informed decision-making. We recommend that health agencies consider incorporating such hybrid models into their early warning systems and policy planning tools. This could support more accurate forecasting, resource allocation, and intervention strategies during outbreaks. Future research should focus on extending this approach to other infectious diseases, evaluating model generalizability across different populations and settings, and optimizing computational efficiency to support real-time deployment in diverse public health environments.

Supplemental Information

Supplemental Information 1 Algorithms and code.

Supplemental Information 2 ARIMA Parameter Selection and Model Performance of Active Cases.

Supplemental Information 3 ARIMA Parameter Selection and Model Performance of Recovery Cases.

Supplemental Information 4 ARIMA Parameter Selection and Model Performance of Death Cases.

Supplemental Information 5 LSTM Model parameter for Active, Recovery, and Death cases.

Supplemental Information 6 Diebold-Mariano (DM) Test Results Comparing Forecast Accuracy Between Models.

The authors would like to express their sincere gratitude to all individuals and institutions who supported this research. We also appreciate the technical and institutional support provided by Universiti Sains Malaysia. In addition, we acknowledge the use of AI-based editorial tools, including Grammarly and Write full, for language refinement, grammar correction, and maintaining an academic tone during manuscript preparation.

Additional Information and Declarations

Competing Interests

The authors declare that they have no competing interests.

Author Contributions

Al Mahmud conceived and designed the experiments, performed the experiments, analyzed the data, performed the computation work, prepared figures and/or tables, authored or reviewed drafts of the article, and approved the final draft.

Syed Husni Noor Syed Hatim Noor performed the experiments, analyzed the data, performed the computation work, authored or reviewed drafts of the article, and approved the final draft.

Kamarul Imran Musa conceived and designed the experiments, performed the experiments, analyzed the data, performed the computation work, authored or reviewed drafts of the article, and approved the final draft.

Firdaus Mohamad Hamzah conceived and designed the experiments, performed the experiments, analyzed the data, performed the computation work, authored or reviewed drafts of the article, and approved the final draft.

Zainab Mat Yudin conceived and designed the experiments, performed the experiments, prepared figures and/or tables, authored or reviewed drafts of the article, and approved the final draft.

Noorshaida Kamaruddin conceived and designed the experiments, prepared figures and/or tables, authored or reviewed drafts of the article, and approved the final draft.

Ashwini M. Madawana conceived and designed the experiments, prepared figures and/or tables, authored or reviewed drafts of the article, and approved the final draft.

Mohamad Arif Awang Nawi performed the experiments, analyzed the data, performed the computation work, authored or reviewed drafts of the article, and approved the final draft.

Data Availability

The following information was supplied regarding data availability:

For full reproducibility, the implementation scripts are available at Zenodo:

Awang-nawi. (2025). Awang-nawi/AI_COVID19_Forecast_Models: Hybrid ARIMA-LSTM (v1.0.0). Zenodo. https://doi.org/10.5281/zenodo.16901882.

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
