# Peer review of "Hybrid ARIMA-LSTM for COVID-19 forecasting: a comparative AI modeling study"

_PeerJ Computer Science, doi:10.7717/peerj-cs.3195_

## Round 0.1 · original submission · Major Revisions

Dear Authors,
Thank you for the new version of your contribution.

After carefully evaluation the reviewers' reports for this version, I have to inform you that some issues need to be addressed.

The reviewers acknowledge the relevance of your study on hybrid ARIMA-LSTM models for COVID-19 forecasting, particularly in a localized context. However, several important issues must be addressed to improve the clarity, rigor, and impact of your manuscript.

The text seems to contain numerous language and grammatical issues, including inconsistent terminology and typographical errors. Reviewers strongly recommend a complete language revision. The dataset usage appears to be inconsistently described, with unclear justification for the omitted data ranges and an overly small test set that compromises statistical validity. A more comprehensive use of the dataset and a larger prediction window are recommended.

The methodology section lacks sufficient detail on model integration and hyperparameter tuning. Key steps in data preprocessing, such as handling missing data or performing stationarity checks for ARIMA, are not described. Diagrams or pseudocode would aid in making your workflow reproducible. The GitHub repository is incomplete (missing scripts, insufficient documentation) and should be updated accordingly.

Figures and tables require more transparent labeling, improved clarity, and higher resolution. Captions should fully explain model distinctions, axes, and color schemes. The literature review is outdated and thin, weakening your study's position within the current research landscape. Please update it with recent sources and more clearly emphasize the novelty of your study.

Finally, reviewers request statistical significance testing for model comparisons and suggest discussing external factors (e.g., lockdowns, vaccination rates) that may affect model accuracy. Consider revising the title for clarity and impact, and expand the conclusion to include practical recommendations.

**Language Note:** The Academic Editor has identified that the English language must be improved. PeerJ can provide language editing services - please contact us at [email protected] for pricing (be sure to provide your manuscript number and title). Alternatively, you should make your own arrangements to improve the language quality and provide details in your response letter. – PeerJ Staff

·

Basic reporting

1. In Line 147, the data from July 18, 2021 to August 14, 2021 is said to be used for testing, which contradicts statement in Line 228 where it is mentioned that the same data used for validation
2. In Line 251 and Table 2, evaluation_metrics.py is mentioned but it doesn’t exist in the github link https://github.com/Awang-nawi/AI_COVID19_Forecast_Models
3. Grammar error in Line 139 - Consider changing “describe detail as Fig. 1 below” to “described in detail in Fig. 1 below”
4. Consider changing Lines 144, 202, 227, 272, Figure 1 and Figure 2 from “time series” to “time-series”
5. Typo in Line 156 - change “phyton” to “python”
6. There’s a lot of inconsistent naming usage for the hybrid model. Please use consistent ordering, capitalization and hyphen usage. For example in in lines 162,168,192,195,223, 264 and 278 the usages vary between “Hybrid ARIMA LSTM”, “ARIMA LSTM hybrid”, “Hybrid ARIMA-LSTM”, “ARIMA-LSTM hybrid”, “hybrid ARIMA-LSTM “, “ARIMA-LSTM (Hybrid)”, “ARIMA-LSTM Hybrid”
7. Add spacing in Line 226 - change “(9,2,2)” to “(9, 2, 2)”
8. In Figure 2, the "Validation Process" block mentions only 3 metrics. Why not all 5?
9. In Figure 5, the title is abruptly cut out in the picture

Experimental design

1. To provide conclusive statistical evidence, the authors should consider predicting on a larger sample set. Using 7 values for prediction is not enough data for statistical significance. Even with a modest 75% train, 15% validation, 15% test split on the provided dataset, the authors should have a prediction sample size of greater than 1 month.
2. Why wasn’t the full dataset utilized? For example from Lines 227-229, data between Aug 15 to Sept 11 and Sept 19 to Sept 24 is neither used in training, validation or testing

Validity of the findings

no comment

Cite this review as

Reviewer 2 ·

Basic reporting

1. English Language and Grammar
The document uses normally understandable English but often suffers from grammatical issues and awkward sentence structures that impede clarity. For example:
-Line 131: "COVID nineteen daily data" should read as "COVID-19 daily data."
-Line 156: "The model is trained using phyton software" → Must be altered to "Python software."
-There are also article usage issues ("the", "a", "an"), plural, and verb conjugation issues throughout.
-Recommendation: Edit by a native English speaker or a professional editing firm to improve sentence structure, fluidity, and scientific tone.

2. Organization and Layout
-The manuscript is standard IMRAD (Introduction, Methods, Results, and Discussion) in organization but has some redundant or too lengthy sections.
-Example: The introduction recapitulates the same sentiments about model limitations in multiple paragraphs.
-Recommendation: Cut redundancy and improve transition between paragraphs for improved logical flow.

3. Figures and Tables
Figures 3–5 need more effective labeling:
-Captions need to define all model lines (e.g., ARIMA in green, LSTM in red) precisely.
-Axes labels do not appear in figure descriptions. Figures should be self-explanatory.
-Table 3 is overpopulated and may be difficult to read for some.
-Suggest: Split it into three subtables (cases, deaths, recoveries) or expand visual spacing.

4. Background and Literature Review
-Literature citations are proper, yet some of them are older or over-referenced (e.g., Box et al., 2015; Hochreiter & Schmidhuber, 1997).
-The comparison between LSTM, ARIMA, and hybrid models is good, but novelty for the study can be better established relative to existing literature.
-Recommendation: Include 1–2 recent citations (2022–2024) on hybrid models in pandemic prediction to establish the contemporary relevance of your study.

5. Terminology and Consistency
There are discrepancies in the spelling and case:
-"Covid-19 recoveries," "covid-19 deaths," and "COVID-19 cases" vary in capitalization.
-Use the same model name case consistently (e.g., always "ARIMA-LSTM").
-Recommendation: Standardize the scientific terms, acronyms, and abbreviations in the manuscript.

6. Raw Data and Code Transparency
The manuscript appropriately includes a GitHub link to datasets and code.
-But no information on the format of the dataset or how to run the code provided is given.

Experimental design

1. Methodological Clarity:
The description of the hybrid ARIMA-LSTM model lacks sufficient detail on how residuals are transformed and fed into the LSTM. A schematic diagram or pseudocode would significantly improve reproducibility.

2. Hyperparameter Tuning:
Although the paper mentions "iterative experimentation," the tuning process is vague. Authors should include more details about how hyperparameter values (e.g., batch size, number of units, epochs) were selected.

3. Data Preprocessing:
While normalization and train-test splitting are mentioned, there is no discussion of missing values, outlier detection, or stationarity checks for ARIMA. Please provide this to validate the robustness of preprocessing steps.

4. Reproducibility:

While the GitHub link is provided, ensure that the README file clearly describes dependencies, preprocessing scripts, and model usage to enable complete reproduction.

Validity of the findings

1. Model Comparison:
The performance comparison between ARIMA, LSTM, and ARIMA-LSTM is well presented, but statistical significance testing (e.g., paired t-test or Wilcoxon test) should be included to confirm that performance differences are meaningful.

2. Limitations and Generalizability:
The authors acknowledge the study is limited to Malaysian COVID-19 data, but they do not test generalizability on other datasets. While not mandatory, using another dataset (even a smaller one) would significantly strengthen the findings.

3. Forecasting Horizon Justification:
The justification for selecting the test and validation periods is unclear. Why was September 12–18 chosen? How does this week reflect model performance under realistic pandemic conditions?

4. Impact of External Factors:
The manuscript does not consider the impact of external variables such as lockdowns or vaccination rates. Authors should either incorporate such variables or discuss how their omission may bias model accuracy.

Additional comments

1. Contribution and Novelty:
The study provides a useful comparison of classical and hybrid AI models for forecasting pandemics, and the integration of ARIMA with LSTM is timely. However, this concept has been explored before. Emphasize more clearly what sets this study apart (e.g., empirical evidence on Malaysian data, comparative evaluation).

2. Title Refinement:
The current title is informative but could be shortened for impact. Consider: “Hybrid ARIMA-LSTM for COVID-19 Forecasting: A Comparative AI Modeling Study.”

3. Conclusion Expansion:
The conclusion should provide more actionable insights (e.g., model deployment feasibility, computational considerations, or recommendations for health agencies).

4. Technical Enhancements:
Provide equations for the LSTM architecture or flowchart of the hybrid model for readers unfamiliar with these methods.

Cite this review as

·

Basic reporting

The subject of this article is of great importance, particularly in light of recent global health crises. The integration of classical statistical models (ARIMA) with advanced deep learning models (LSTM) represents a valuable methodological approach, offering enhanced prediction accuracy by capturing both linear trends and non-linear patterns within epidemiological data.
The language used throughout the article is generally clear and accessible; however, there are a number of linguistic and technical issues that require correction. For example, in line 123, the sentence “data will be collected from the official” is incomplete and unclear, data already collected. Similarly, in line 156, the phrase “The model is trained using phyton software” contains a typographical error (“phyton” instead of “Python”) and a vague technical description. A more accurate phrasing would be: “The model is trained within a Python environment using relevant machine learning libraries.” Such issues point to the need for a thorough language review to enhance overall clarity and professionalism.
Another limitation lies in the article’s literature review. It references only 11 sources, with the most recent being from 2021. Considering the paper is presented in 2025, this significantly weakens the research’s foundation. Several recent and relevant studies have been omitted, including a pertinent article published in the International Journal of Environmental Research and Public Health: https://www.mdpi.com/1660-4601/22/4/562, which provides valuable insights into pandemic forecasting methodologies. An updated and expanded literature review is necessary to position the research within the current academic discourse.
In addition, Figure 1 is unclear and lacks sufficient resolution or explanatory detail.

Experimental design

The choice of dataset could also be strengthened. While the study uses a local dataset, broader and more robust data sources, such as the Covid-19 database from Johns Hopkins University’s CSSE, should have been considered. This dataset includes comprehensive global data including for Malaysia across a substantial time span, thereby offering a more rigorous foundation for model training and testing.
Lastly, the article’s evaluation metrics are rather limited. For a more holistic assessment of model performance, the authors should incorporate additional metrics such as Normalised Root Mean Squared Error (NRMSE) and the Coefficient of Determination (R²). These would provide a more nuanced understanding of both the model's predictive accuracy and its ability to generalise.

Validity of the findings

The novelty and contribution of the study are not sufficiently substantiated. The results, although briefly presented, are not effectively contextualised through comparative analysis with existing work. A more rigorous benchmarking against similar studies. Furthermore, the study would benefit from the inclusion of additional figures and diagrams to better illustrate the methodology and outcomes. An updated and expanded literature review is necessary to position the research within the current academic discourse.

---

## Round 0.2 · Major Revisions

Dear Authors,

Your approach is interesting, and the reviewers thank you for addressing all their concerns. However, I have some other concerns that I have to request you address. As your manuscript contains a statistical analysis, I believe the methodology must be documented and validated rigorously.

Please, take into account the following concerns.

Please clarify the choice of the ARIMA model order (p = 9, d = 2, q = 2). It is hard to understand the supporting evidence. It would be helpful to discuss and present information-criterion comparisons across a grid of candidate (p, q) values.

Also please discuss residual diagnosis and verify if white noise and errors are independent or have behaved such as identically distributed.

You may consider improving your validation strategy which, if I understood well, relies on a single train/validation/test split, which risks overfitting the results to one arbitrary partition.

Finally, although you report seven performance metrics, I am unsure regarding confidence intervals or formal statistical tests, and whether you can accompany the claim that the hybrid model performs “best”. Maybe you can try to assess whether the observed differences in forecast errors between models are statistically significant. In addition, several models exhibit negative R² values but I was unable to understand where you discuss the interpretation. A negative R² indicates that the model performs worse than simply using the historical mean as a predictor; this should be explicitly explained.

·

Basic reporting

no comment

Experimental design

no comment

Validity of the findings

no comment

Cite this review as

·

Basic reporting

I am happy with the new submission. I appreciate the authors' thorough revision in response to the previous comments; the core manuscript improvements are evident. My primary remaining concern relates to referencing accuracy. Please carefully review the reference list for correctness (including author names and links/DOIs) and ensure consistency between all in-text citations and the list entries. Once these referencing issues are resolved, the submission will be strong.

Experimental design

Good

Validity of the findings

Good

Additional comments

all good

---

## Round 0.3 · accepted · Accept

Dear authors, thank you for addressing my concerns.